# Dilatation of New Progressive Hybrid Sand and Its Effect on Surface Structure, Roughness, and Veining Creation within Grey Cast Iron

**DOI:** 10.3390/ma16052004

**Published:** 2023-02-28

**Authors:** Martina Bašistová, Filip Radkovský, Ivana Kroupová, Petr Lichý

**Affiliations:** Department of Metallurgical Technologies, Faculty of Materials Science and Technology, VSB-Technical University of Ostrava, 17. Listopadu 2172/15, 708 00 Ostrava, Czech Republic

**Keywords:** thermal expansion, foundry moulding mixture, silica sands, CERABEADS, veining, surface defects, artificial ceramic sands, surface structure, surface roughness

## Abstract

The constant effort of all metal alloy manufacturing technologies and processes is to improve the resulting quality of the processed part. Not only the metallographic structure of the material is monitored, but also the final quality of the cast surface. In foundry technologies, in addition to the quality of the liquid metal, external influences, such as the behaviour of the mould or core material, significantly affect the cast surface quality. As the core is heated during casting, the resulting dilatations often lead to significant volume changes causing stress foundry defects such as veining, penetration and surface roughness. In the experiment, various amounts of silica sand were replaced with artificial sand and a significant reduction in dilation and pitting of up to 52.9% was observed. An important finding was the effect of the granulometric composition and grain size of the sand on the formation of surface defects from brake thermal stresses. The specific mixture composition can be considered as an effective prevention against the formation of defects instead of using a protective coating.

## 1. Introduction

The everlasting development of manufacturing methods of steel and cast iron gave rise to modern processes, such as powder metallurgy [1], or additive manufacturing [2]. Nevertheless, these methods feature typical disadvantages, such as the presence of porosity [3] or inhomogeneous distribution of residual stress [4], and are still being developed. Also by this reason, conventional casting and casting-based methods are still widely used in the industry, although they also feature certain downsides.

In conventional casting, uneven temperature distribution occurs in the cross-section of the sand mould due to the different distance from the mould-metal contact surface. Due to this phenomenon, the mould expands unevenly and stresses are generated, whereby cracking occurs when the ultimate strength of the mould surface or core in contact with the molten metal is exceeded [5,6]. This allows the liquid metal to penetrate the resulting gaps/slots. The high fluidity of foundry alloys exponentiate this crack filling [7], resulting in defects such as veining, rat tails and, to a lesser extent, increased roughness of the cast surface.

Veining are most often found on the rounded edges and cylindrical surfaces of the casting in the form of fine protruding veins, ribs or nets and are a typical surface defect resulting from dilatation and stress of the moulding compounds [8]. Those defects can becommonly found in castings made from alloys with high fluidity (ferrous alloys, some non-ferrous metals) that are cast in moulds made from silica sands. In the case of steel castings, we are more likely to encounter veining accompanying strong penetration. In most cases, they are easy to remove, but removing them lengthens production and increases costs. However, in hard-to-reach parts of the castings (cavities), they can also completely invalidate the casting [9,10]. Increased surface roughness is also an equally important defect; for example, in a study [11] investigating the effect of roughness in cast iron castings on fatigue strength, it was confirmed that the test specimen with lower roughness showed higher fatigue strength.

The thermal expansion of foundry sand largely influences the mechanism of the formation of the veining and penetration type defect. The most commonly used sand in the foundry industry, for economic reasons, is silica sand, which is also the most readily available and meets the requirement for high refractoriness of the moulding compound. Its biggest disadvantage is its characteristic discontinuous thermal expansion at elevated temperatures accompanied by a high-volume change of up to 3.9%. This phenomenon occurs during the thermal phase transition of SiO_2_ already at a temperature of about 573°C [12,13,14]. For high-purity silica sand in the presence of mineralizers (K, Na ions from water glass, alkaline resin catalysts, alkalized resol) exposed to high temperatures (iron alloy casting), dilatation and stresses further increase along with the possibility of SiO_2_ conversion to cristobalite, which is accompanied by an additional volume change of up to 15%. [9,15,16]. The macro-dilation of the mould or core is then dependent on the micro-dilation of each grain, which is influenced by other parameters such as: the composition of the sand and grain shape, the type and content of the binder, the additives used, the degree of compaction and others. The highest value of dilatation is observed in monofractional sands with rounded grains [14,17,18]. Testing of different types of sands and their propensity to form surface defects in castings has been previously addressed by various authors [8,19,20,21], where most theories are based on the thermal stresses in silica sands, mainly because no veining is observed in non-silica sands (chamotte, corundum, Cerabeads), as they do not undergo discontinuous dilation.

As a measure to prevent the formation of veining and to improve the overall quality of the cast surface, the origin of the silica sands and its chemical purity can be changed [20], or the granulometric composition can be changed [22]. By replacing the silica sands or part of it with non- silica sands, we can achieve an increase in the refractoriness of the moulding compound together with a reduction in stress (dilatation) [9,23,24]. Another possibility is also the use of relaxing additives (expanded perlite), which after thermodestruction (burn-out) will release the intergrain spaces and allow the grains to expand [25]. Another option is the addition of iron oxide to the forming mixture to form low-melting silicates (fayalites) and increase protection against metal penetration and vein type defects [26]. The effect of mineral additives has been investigated by the authors [27] to assess the effect on extensibility and reduction of veining. Other authors [28], on the other hand, state that fluxes can also be used to reduce the occurrence of veining, which help to create a solid surface and thus prevent the penetration of liquid metal into the moulding mixture. These defects have also been observed in thin-walled castings, where, for example, the authors of [29] introduced new additives to eliminate veining and penetration defects. However, the most common method remains the application of various protective, penetrating or graphite coatings [7,30].

The identification of veining or roughness type defects is mainly done visually on the final casting or by microscopes [31,32], but there are already published studies where the propensity for veining type defects and metal penetration into the mould were tested by simulation. For example, the effect of changing variables in the manufacturing process [33] or the effect of a new additive on reducing metal penetration has been studied [34]. Another study by the authors [18] shows the possibility of successfully designing a cracking criterion using maximum tensile stress and effective volume, where the accuracy of the cracking criterion was confirmed by a series of tensile tests and simulations. This enabled them to determine the potential for mould and core cracking during casting prior to production. Equally important are casting tests [28], where the propensity for surface defects can be observed and tested. By keeping the foundry parameters identical, it is possible to obtain results helpful to understand the influence of the different materials used.

The formation of defects due to thermal stresses in moulds and cores represents a high time and financial cost for the foundry. In the case of small defects on the accessible area, there is a need to repair the surface by grinding, in the case of defects in inaccessible areas or internal cavities, it is a scrap casting. As has already been mentioned, it is more than desirable to prevent these defects from occurring. The most common prevention is the use of protective coatings, especially on cores. However, the application of coatings requires a certain degree of specialisation, either by investment in equipment for applying the coating (pouring, spraying, dipping) or, in the case of manual application, manual skills. It is the incorrect application of the coating that is often the reason for the failure of this prevention. The aim of this experiment was to define a possible protection against the formation of surface defects such as veining by modifying the composition of the moulding mixture, which would enhance the protective effect of the coating or ideally be able to replace it completely. This possibility has not yet been studied more and was not fully exploited in foundries. Different compositions of the hybrid moulding compound were investigated, with silica sand gradually being replaced by Cerabeads artificial sand, up to 50% of the content due to the high purchase price. The granulometric composition and its effect on sand dilatation also was evaluated. A gradual reduction in the rate of dilatation and avoidance of veining, penetration or increased surface roughness accompanying thermal stress defects were expected.

## 2. Materials and Methods

A simple schematic representation of the sequence of individual measurements and sample production can be seen in Figure 1. A description of the individual measurements, including the material used, is defined in the following chapters.

### 2.1. Selection and Evaluation of Sand Properties

Silica sands for foundry or glass making purposes are in high quality, generally with a SiO_2_ content of at least 95% [35,36,37]. For the purpose of evaluation of the influence of thermal expansion of the used hybrid sand on the final surface quality of the iron alloy casting, the Polish silica sand from Biala Góra (supplier mark BG27 [38]) with purity < 99.7% SiO_2_ (declared by supplier and confirmed by study [39]) and medium grain size AFS 60 was selected as the initial sand. For the admixture, CERABEADS (hereafter CB, Japan origin) artificial aluminosilicate round grain sand was selected in three grain sizes (designations CB-65, CB-95 and CB-145) ranging from coarse grain, comparable to the silica sand used, to very fine grain suitable for special applications. The raw materials and characteristics used are based on a previous experiment evaluating the basic properties of the mixture. All sands used are characterised by similar bulk density (performed on tapped samples by vibration with results 1.66–1.76 g/cm^3^), which guarantees good miscibility without dilution [24].

The hybrid mixture was created by gradually replacing part of the silica sand with artificial Cerabeads (CB) in a content of 20–50%. The content of 10% CB addition was now omitted from the experiment due to the expected minimal effect on the hybrid mixture properties and the resulting casting surface quality. Contents higher than 50% were also not included due to the high purchase price of Cerabeads, and thus the economic inefficiency of the hybrid blend. A total of 12 hybrid sand mixtures were prepared in this way, 1 pure reference sample containing 100% BG27 silica sand and also 3 samples containing 100% CB-65, CB-95 and CB-145 to monitor the change in granulometric composition. The marking of the samples and their composition can be seen in Table 1. This marking was also used to mark the castings where these mixtures were used.

First, the granulometric composition of the hybrid mixture was determined based on its composition. Since the CB species differ in mean grain size, it was necessary to define exactly how their admixture would affect the overall composition of the sand mixture in terms of particle size refinement and change in the character of the grading of the sand (monofraction, polyfraction). In order to evaluate the granulometric composition, sieve analysis including determination of washed out particles according to ASTM E11 was performed using a laboratory sieving machine (Multiserw-Morek, LPzE-2e, Poland) and sieves with mesh size 0.063–0.710 mm. The percent of sand fraction captured on the nets is expressed as the amount of sand captured on the nets to the total weight of the sample. The sieving time was set at 10 min. From the results, cumulative curves of granularity were then constructed and the log W (Criterion of the grain size distribution probability) and AFS mean grain size were read.

The thermal expansion of the hybrid sand mixtures and pure reference samples was measured on a DIL 402/C dilatometer (Netzsch, Germany). This dilatometer is equipped with corundum components including a corundum container with plugs for the measurement of bulk materials. Calibration was performed using a Ø6 × 10 mm Al_2_O_3_ correction sample. The individual measured samples of the hybrid mixtures were poured into the container and homogenized by means of laboratory rod strikes. The sample height was varied within 10 ± 0.05 mm. An inert atmosphere of 6.0 argon (100 mL/min) was provided for the measurements and a temperature range of 25–1130 °C with a temperature rise of 15 °C/min was selected.

Three samples from each hybrid mixture were evaluated each time for both sieve analysis and dilatation, and the resulting values were averaged.

### 2.2. Core Production and Determination of Bending Strength

The 50 × Ø50 cores were formed by Cold-box technology using 0.8% Leganol binder, mixing was performed in a batch mixer for 1 min. The curing was carried out on a laboratory injection moulding machine equipped with a metal corebox using tertiary amine DMIPA. For core production and subsequent evaluation of the surface quality of the casting, mixtures with 20–50% addition of CB-65, CB-95 and CB-145 were selected based on the results of thermal dilation. To compare the surface quality results, test cores were made from 100% pure silica sand BG27, which has a high potential for veining and surface roughness and thus serves as a reference sample. An alumino-silicate alcohol coating was applied to a portion of the 100% BG27 cores, which is commonly used as a common protection against veining and to improve the overall surface quality of the cast material. The surface roughness investigated was assessed in terms of stress defects caused by thermal expansion of the sand, where increased roughness can be the mildest manifestation and often accompanies veining. On the other hand, increased surface roughness can also be a manifestation of too coarse-grained sand, sparse (i.e., porous) cores, with liquid metal penetrating the intergranular spaces. Therefore, efforts were made to minimize this cause by producing cores by ramming into metal cores, whereby a highly dense and smooth surface layer was achieved. Also, the use of a protective coating on the reference core served as protection against metal penetration into the sand and to increase the smoothness of the cast surface.

The production and use of 100% CB cores were not resorted to as they are characterized by almost zero occurrence of stress defects on castings.

Determination of 3-point bending strength was carried out on test beams with dimensions 22.5 × 22.5 × 170 mm made of the above binder and hybrid sand cured in metal corebox on the LRu-2e measuring instrument (Multiserw-Morek, Poland). Measurements were carried out immediately after curing and 24 h after curing on at least 3 samples each time.

### 2.3. Preparation of Test Castings for Casting Test

The model of the test casting for the occurrence of veins was made according to Figure 2 (own). The made model was modified by removing the sprue system due to the need to reduce the weight of the insert because of the limit of the melting aggregate.

The mould was made by hand from a bentonite moulding compound consisting of 100 wt.% silica sand, 8 wt.% bentonite and 2 wt.% carbonaceous additives. Three cores with the same content of each CB (e.g., 50% CB-65, 50% CB-95 and 50% CB-145) and one of 100% BG27 coated with or without aluminosilicate coating, as a reference sample.

The casting material was an iron alloy, namely grey cast iron EN-GJL-200, because of its high fluidity and high casting temperature, which significantly exceeds the modification temperature of β-SiO_2_ to α-SiO_2_, thus also increasing the probability of stress defects. The chemical composition of the cast iron is not prescribed by the standard, it is individually governed by the wall thickness and the prescribed mechanical properties according to EN 1651:2011. The casting temperature was set at 1430 °C and the casting time was 18 s. The casting was filled through the cavity of an atmospheric feeder located in the center of the casting. The gross weight of the casting was 27 kg. Two castings were made for each hybrid mixture sample.

### 2.4. Casting Surface Quality Evaluation

The evaluation of the surface quality of the casting surface focuses mainly on the existence of stress defects, namely, veining. As a complement, the experiment includes a surface roughness measurement.

The casting removed from the mould was cleaned of any residue of the moulding compound. The cores were then carefully cleaned with a needle so as not to disturb the veining present in the cavity. The width and length of the pits that were present in the core were then measured. From these values, the dimension of the area in which the protuberances were present was obtained, and this was then expressed as a percentage. Areas with other defects such as metal penetration were not included in the measurements. These were simply identified by the nature of the defects, with the veining forming distinct veins on the surface of the casting.

The cleaned cast, after measuring the occurrence of the projections, was then cut into four quarters so that each quarter of the cast contained the entire examined cavity arising from the entire founded core. This was then subsequently divided into further approximately equal quarters across the center of the cylindrical cavity so that they could be placed under the optics of the microscope. The surfaces of the individual quarters were cleaned with a steel brush to remove any residual sand grains. The roughness of the cast surface was measured on the specimens thus prepared, the roughness values always being read in the areas near defects (if possible) and from the center of the surface, where 5 separate measurements were performed. The center of the casting was chosen with regard to the possible presence of impurities in the upper and lower parts of the casting and also because of the possible increased occurrence of baked sand near the detected defects. In case of error measurement, the comparative measurement was done. Thus, at least 20 values were obtained from each cavity along the core. The results were then averaged. In case a veining was present in the selected location for measurement, the roughness measurement was performed on the nearest surface next to the veining. Measurements were performed on a VHX-6000 digital optical microscope (Japan, Keyence) at 500× magnification.

## 3. Results

### 3.1. Evaluation of the Properties of Hybrid Sands

Based on the sieve analysis of the individual hybrid sand mixtures, it was possible to precisely define the granulometric composition of the individual samples and thus determine the degree of influence of the resulting mixture by mixing silica sand BG27 with artificial sand CB. From the measured data it was possible to evaluate the change in mean grain size AFS and to determine the log W value, which gives us an indication of the nature of the sand sorting. The measured results for the sequentially blended mixtures CB-65, CB-95 and CB-145 including the reference value for 100% BG27 silica sand can be seen in Table 2.

If we compared all of the used starting sands at 100% content with each other, there would be a noticeable difference in the mean grain size by AFS between the samples, with the coarsest for the BG27 silica sand AFS 61 and the finest AFS 98 for CB-145. The results of the sieve analysis also show a difference in the sorting pattern of the different 100% sands. Log W gives us an indication of how close the sand was to monofraction (value closer to 0) or if it was more of a polyfractional sort (value closer to 100) [20]. In practice, this means that the sand was more likely to contain grains of the same size captured on one or two sieves (monofraction), or all sand grain sizes were equally represented and all sieves had more or less the same proportion (polyfraction). It is evident from the sieve analysis that the 100% CB-145 sample was more monofractionated in nature compared to the other CB and BG27 sand samples, with 94% of all sand grains containing only 2 fractions.

The results also show that mixing silica sand with artificial CB in all three cases progressively reduced the mean grain size, thus refining the resulting hybrid mixture. The greatest reduction in grain size from AFS 61 to AFS 81, i.e., 30%, occurred in the case of mixing BG27 with 50% CB-145. Also, the character of the sorting defined by log W was different from the reference 100% samples and all hybrid mixtures became polyfractionated. In the case of any addition of CB-65, there was only a minimal change in log W compared to the baseline 100% BG27. In the case of finer CB-95 and CB-145 in any volume of blending, there was an increase in polyfractance compared to the baseline 100% BG27, up to 11%, due to the addition of finer particle amounts compared to the reference sample.

Thermal expansion measurements were performed for all hybrid mix samples with 20–50% CB content as well as for 100% reference samples of BG27 silica sand and CERABEADS artificial sand. The results of the measurements can be seen in Table 3. The dilatation progress can be observed from the dilatation curves in Figure 3. The reference sample of pure silica sand at 100% BG27 content achieved a dilatation of 1.48% with a significantly non-fluid thermal dilatation curve. The phase transformation, typical of SiO_2_, occurred at 554.40 °C, which was accompanied by a characteristic abrupt volume change. In contrast, the 100% CB reference samples achieved up to 4 times lower dilation values, which ranged from 0.36–0.56% with a gradual linear progression. In the case of the hybrid sand mixes, a sharp decrease in dilation was observed only in the case of the addition of 20% CB-65, to a value of 1.28%. In the case of the other CBs tested at the same content, the resulting dilatation was not as significantly affected and specifically in the case of the addition of CB-145, the effect was almost negligible. On the other hand, the highest influence, and therefore the lowest measured dilation, was achieved with the addition of 50% CB-145, namely a value of 1.01%. From the time of the phase transformation around 573 °C until the end of the measurement, i.e., 1130 °C, the measured dilatation remained at a similar level. However, the abruptness of the volume change during phase transformation was more pronounced than in the case of the addition of 50% CB-65, and for which the volume change during phase transformation was less abrupt.

With the gradual addition of CB-95, although the stresses decreased, the steepness of the volume change (steeper dilation curve with a steep increase from the beginning) increased during the β↔α SiO_2_ conversion around 573 °C for the 30% and 40% CB-95 samples, as shown in the dilation graph (Figure 3). The 50% CB-95 sample exhibited the most gradual volume change, despite the fact that the resulting dilation value was almost identical to the 30% and 40% CB-95 samples. This is explained by the suppression of the typical manifestation of high SiO_2_ content under thermal loading, i.e., abrupt and sudden volume change, by the high degree of its replacement by sand with low dilatation. Here, although dilation was higher at low temperatures, it was gradual (more linear curve) and the volume change during phase transformation was not as high.

The summary can be as follows: a coarser grain was expected to dilate more. On the contrary, the influence of monofraction sand sorting CB-145 achieved higher expansions in the clean state than CB-95. In the case of hybrid mixtures, this trend remained with 50% CB-145, when the total dilation was the lowest, but the volume change was the highest and most rapid of the 50% CB mixtures.

### 3.2. Evaluation of Core Strength

Test beams with dimensions 22.5 × 22.5 × 170 mm were made from the selected hybrid mixtures and a reference sample of 100% BG27 silica sand and Cold-box binder system (Leganol binder) to determine the 3-point bending strength of the mixtures. Samples containing 100% CB were no longer included in this measurement. The results of the measurements immediately after curing and after 24 h can be seen in Table 4. The fabricated beams were stored in laboratory conditions at a constant temperature of 23 °C and humidity of 20% for 24 h until the second measurement. It can be noted that immediately after curing, none of the hybrid mixtures achieved the strength characteristics of 100% BG27. After 24 h from curing, an increase in strength could be observed for all mixtures. In the case of the strength results immediately after curing, the strength of the hybrid mixtures with the same CB was either at a similar value or increased very slightly with increasing content. The smallest increase in strength between 0–24 h after curing was observed for the 100% BG27 reference sample, at only 13.0%. On the other hand, the largest increase in strength was in the case of 20% CB-95, namely 57.4%. For all the hybrid mixtures, it was observed that the highest increase in 3-point bending strength occurred in the case of 20% CB blending and as the percentage of CB increased up to 50%, the increase in strength between 0–24 h decreased. Overall, it can be concluded that there was a slight decrease in the strengths of the hybrid mixture at 24 h after curing with increasing CB content due to changes in the granulometric composition of the hybrid mixture.

The summary can be as follows: the CB-65 mixtures showed the highest strengths both immediately after curing and after 24 h (in average 4.6 MPa), without significant differences in the addition of CB. with CB-95, with the gradual addition of CB, the strength decreases after 24 h (17% lower strength in case 50% CB-95 compared to 20%). CB-145 achieves lower strengths than CB-65 (in average 3.65 MPa), but their levels do not change significantly with the addition of CB.

### 3.3. Casting Test Results and Casting Surface Quality

After the casting test and cleaning of the samples of castings made of grey cast iron, the evaluation of the area of castings affected by the veining was carried out. The blemish type defects are easily recognizable by their character of protruding veins and nets, so that they were not confused with other occurring defects (e.g., penetration or burnt sand). Areas containing defects other than veining were omitted from the assessment. The dimensions of the embedding cylindrical core made of the hybrid mixture were Ø 50 × 50 mm. The surface area of the casting formed by these cores was calculated to be 8247 mm^2^. The dimensions of each found core were measured and the result averaged from both test castings. Next, the resulting cast surface roughness Ra was evaluated for the prepared casting samples as a function of the hybrid compound used. The occurrence of veining and the cast surface roughness of the iron alloy castings can be seen in Table 5. 100% CB samples of all 3 types were not included in the casting tests.

The 100% BG27 reference samples were in some cases treated with a protective alumino-silicate coating, which is a commonly used product in foundries for the purpose of protecting against veining and improving the overall quality of the cast surface. However, the use of coatings requires a specialised workplace to regularly check the quality of the coating, as their inexpert application may lead to the disruption of the surface layer of the mould or core by brushing, the formation of bubbles in the coating layer, the peeling off of too thick a layer or, on the contrary, the application of an insufficient layer, all of which may result in an increased risk of veining and deterioration of the quality of the cast surface. The protection against surface defects in the casting would be enhanced or could be completely replaced by a suitable choice of hybrid sand mixture. Therefore, the difference in the final surface quality of the casting using hybrid sand mixtures compared to the initial reference sample without coating and subsequently with coating was primarily investigated.

Figure 4 shows that in the case of 100% BG27 and the application of the protective coating, there was an 18.1% reduction in the incidence of veining compared to the uncoated sample. This corresponds to the measured lower roughness Ra of the cast surface by 16.2% for the treated sample. Thus, the significant role of the protective coating in providing a higher quality cast surface and reducing the likelihood of veining was confirmed. In the case of the addition of 20% CB-65, both the incidence of veining and the roughness of the cast surface were almost comparable to the 100% BG27 sample with coating, thus compared to the 100% BG27 sample without coating, there was a decrease in the incidence of veining by 16.2% and a decrease in surface roughness by 11.3%. With the further addition of CB-65, there was a gradual decrease in the incidence of veining by a further 18.6% for 40% CB-65, until with the addition of 50% CB-65 the incidence of veining decreased by 52.9% overall. In contrast, however, the cast surface roughness Ra increased significantly, up to 34.8% for the 50% CB-65 sample compared to the 100% BG27 sample, and 60.9% when compared to the 100% BG27 sample with coating.

In the case of the hybrid mixture with CB-95, the incidence of veinings is already reduced by 31.3% in the case of a 20% addition of CB compared to the reference 100% BG27 and by 16.1% compared to BG27 with coating. With increasing CB-95 content, there was an increase in the incidence of veining as expected. The amount of veinings was reduced by only 6.3% compared to 100% BG27 without coating and increased by 15.4% compared to 100% BG27 with coating. However, with the addition of 50% CB-95, the incidence of veining on the casting test was again low compared to the reference sample and the result was almost comparable to 50% CB-65. Despite the sudden low occurrence of veining, the 50% CB-95 sample achieved the highest cast surface roughness of all the samples tested. Overall, there was a 74.7% increase in surface roughness compared to 100% BG27, or 108.4% compared to 100% BG27 with coating. This is a 40% reduction in veining, but a twofold increase in surface roughness compared to standard casting conditions using protective coatings.

The hybrid mixture with the addition of CB-145 always exhibited a high incidence of veining and high roughness of the cast surface compared to other CB types in all samples with different CB contents. Although the resulting values showed some reduction in veining in the addition of 50% CB-145, it was only 7.1% compared to 100% BG27 and 13.5% compared to 100% BG27 with coating, the smallest reduction compared to the other samples. The resulting Ra surface roughness also saw some increase, with a 6.5% increase over 100% BG27, and 27.0% over 100% BG27 with coating. Although this was not the highest value of surface surface roughness measured, combined with the high incidence of veining, it was an undesirable result.

The summary can be as follows: In the case of the CB-65 mixture, the highest reduction in veinings up to 52.9% was achieved with the gradual addition of CB. On the contrary, the roughness gradually increased. Only 20% or 50% of CB-95 addition had the effect of reducing the occurrence of veining, but achieved the highest roughness (up to 74.7%). the addition of CB-145 had no significant effect on the formation of veining, but instead achieved a smoother surface compared to other CBs (only 6.5% higher roughness than BG27).

## 4. Discussion

Based on the evaluation of the hybrid sand mixtures and the casting tests performed, it was possible to evaluate the quality of the cast surface with respect to the occurrence of surface defects such as veining, penetration and increased surface roughness. Generally, in the foundry production when veining, penetration or other stress defects appear, even in a small scale, on the available surfaces, these defects are repaired by extensive grinding works. In that case, the surrounding surface layer and cast structure are removed. If these defects occur in inaccessible areas (internal surfaces of narrow cavities, e.g., valve casting), these defects cannot be effectively removed by grinding and the casting is discarded as a scrap piece. Both cases result in the longer production times and the increased costs. For this reason, understanding the mechanism of formation of these defects is key to minimizing or completely preventing their occurrence. The idea was to support or completely replace the application of a protective coating by creating a hybrid mixture with lower expansion that would help eliminate the formation of veining. Because Cerabeads are among the most expensive foundry sands, completely replacing the entire molding or core mixture is not an economical solution compared to relatively inexpensive coatings. However, the issue of coatings remains the necessity of their application to the surface of the mold as perfectly as possible, which requires different levels of equipment and investment as outlined in this study [30]. Incorrectly chosen coating or its bad or impropriate application can result in the failure of their protective effect as mentioned in the study [40] and result in additional costs or scrapping in due to the formation of surface defects such as veinings.

In view of the above casting test results, the not very visually significant results of lower CB content in the hybrid mixtures and the fact that the greatest effect of reducing the occurrence of veining was achieved by using an addition of 50% CB of all three types, the results of these samples are discussed further in this chapter. Reference samples of 100% BG27 with and without coating are presented for comparison. Figure 5 shows the occurrence of veining and penetration on the samples discussed, where the effect of the hybrid compound on the quality of the cast surface is clearly visible. For each hybrid compound sample, 2 identical castings were made, which showed approximately the same amount and nature of defects.

After cleaning, some of the castings showed additional penetration type defects, namely 100% BG27, 50% CB-95 and 50% CB-145. The most extensive penetration was evident on the 50% CB-95 casting, while the least penetration was evident on the 50% CB-145 casting. On the other hand, the 100% BG27 with coating and 50% CB-65 castings showed no penetration, only veining type defects. No other surface defects were observed on the castings. Penetration was also not evaluated in further detail and was not included in the area of veining as it is a different type of defect.

On 100% BG27 castings, the effect of the alumino-silicate protective coating, which is commonly used as an effective means of preventing the occurrence of veining and penetration defects, is clearly visible. Here, too, there was a significant 18.1% reduction in veining and complete avoidance of penetration. The effectiveness of the use of protective coatings against veining is also confirmed by the results of the casting test in this technical report [41], where, among other things, the effect of the additives of the moulding compound was evaluated. Another effect of the coating effect was to increase the quality and reduce the roughness of the cast surface for the 100% BG27 coating sample, as shown in Figure 6 and the results shown in Table 5, where a 16.2% reduction in Ra roughness was achieved. Thus, when comparing the surface of the two castings using 100% BG27 with and without coating, a greater proportion of trapped (penetrated) sand grains on the surface of the sample without coating is evident, which increased the measured surface roughness. In contrast, the sample with the coating applied showed only grain imprints or small depressions on the surface of the casting. This is due to the covering effect of the coating, which helps the mould face or core against erosion of sand grains and their entrapment in the surface layer of the solidified casting.

In terms of the incidence of veining, the 50% CB-65 hybrid mixture was found to be the most effective, achieving a 42.5% reduction in veining compared to the reference 100% BG27 coated sample and a 52.9% reduction compared to the untreated 100% BG27.This result appears to be a more effective method of protecting the castings from surface defects than the application of a coating system demanding quality of application. Also, visually, the surface appeared more coherent, with only occasional presence of penetrated sand grains or minor depressions. Although the resulting surface roughness increased for the 50% CB-65 samples, it still did not reach the level of the 50% CB-95 samples, and can still be expected to be effective in practice. Although the castings made from 50% CB-95 showed a slightly higher incidence of veining than the 50% CB-65 castings, the Ra surface roughness was 29.5% higher, as evidenced by the surface images in Figure 6, which show the greatly increased roughness due to the imprint of sand grains into the surface metal layer, as well as the strong influence of metal penetration and embedded sand grains, which were most abundant in these samples. The surface roughness and amount of veining in the 50% CB-145 samples was the most similar to the 100% BG27 uncoated reference sample of all the hybrid mixtures tested, with the surface achieving the same characteristics visually.

If we compare the resulting surface quality, the occurrence of veining, penetrated sand and roughness of the cast surface with the results of the granulometric analysis and the reference sample 100% BG27 shown in Figure 7, at first glance a distinctly discontinuous dilatation curve with a sharp volumetric change in the phase transformation temperature accompanied by a high volumetric change is evident. It is this change, characteristic of silica sands, that results in rapid expansion of the cores, cracking them and allowing metal to flow into the resulting voids. If, as in the case of this experiment, an alloy with a high casting temperature and good convergence is used, there is also a high risk of toasting and penetration defects due to the abrupt volume change, the occurrence of which was confirmed in this experiment.

Since finer sands are often chosen to increase the surface quality of the casting, the addition of 50% CB-145 was expected to produce the least defects at the beginning of the experiment due to its fine grain size. The influence of the size of the sand grains on the number of outgrowths is solved, for example, by the authors of this study [13], where coarser quartz sand dilated more than finer sand of the same origin. The magnitude of thermal dilation of silica sand and the temperature of the onset of the β SiO2 ↔ α SiO2 transformation is influenced by its chemical purity, grain shape, grain size and the nature of the sorting, i.e., the tendency to monofraction. Natural silica sands, with their relative proportion of finer and coarser particles, are more akin in character to polyfractionated sands, whereas artificial mixtures tend to be very sharply sorted with a rather monofractionated character, with all grains represented in one or two fractions, i.e., sizes. The magnitude of the resulting thermal dilation of hybrid blends should therefore depend not only on the amount of artificial CB added, but also on its grain size and the nature of the composition. This was demonstrated for the hybrid mixtures tested with the addition of 50% CB, where, in the case of CB-65 (the most common type in the representation of this type of sand), the resulting AFS 64 grain size, distribution and character of gradation were close to the reference sample BG27 with a typical curve for silica sand (Figure 7a). The linear dilatation (Figure 7b), although higher than the CB-145 blend, was much more gradual in the onset of phase transformation and with less volumetric change. Due to its AFS 65 grain size, the cores used achieved higher 3-point bending strengths (Table 4). All of this was then also reflected in the minimal occurrence of veining, and zero penetrations, which outperformed even the commonly used reference sample 1000 BG27 with coating.

Since 100% CB-95 in its pure state is similar in composition to BG27, the resulting hybrid blend with 50% CB-95 mimicked the inverted BG27 reference sample in terms of fractional composition while achieving a finer AFS 77 medium grain. The difference here was that, compared to the reference sample, the blend with 50% CB-95 had a greater representation of grains in finer fractions on the 0.125 mm mesh and smaller, whereas in the case of BG27 this amount was represented on the 0.250 mm mesh (Table 2). The resulting dilatation of this blend then followed the shape of the 50% CB-65 dilatation curve (Figure 7b), but achieved higher values of both linear dilatation and volume change during phase transformation, despite the fact that the mean grain size of AFS 77 was smaller than that of 50% CB-65, and so there was a prediction of lower dilatation. This was again due to the mixture composition, with a large representation of fine grains and proportions in the hybrid mixture filling the inter-grain spaces and preventing the relaxation of thermal expansion, which externally manifested itself in greater linear dilation. Also, the 3-point bending strength at 24 h after curing was 18.5% lower than that of 50% CB-65 (Table 4). This is due to the fact that smaller sand grains exhibit a larger specific surface area which consumes more binder. Since the dosage of binder was always the same in the case of specimen fabrication, the lack of binder resulted in reduced strength and a greater risk to erosion of the core surface or reduced strength of the binder bridges on the core surface during thermal exposure during casting. This was also subsequently confirmed by the casting test, where the 50% CB-95 samples, although showing a similar occurrence of veining as the 50% CB-65 samples, also showed the highest proportion of penetration and the highest proportion of cast surface roughness and baked sand grains (Figure 6), precisely because of the reduced core strength caused by the unsuitable granulometric composition.

The 100% pure CB-145 sample was characterized by its smallest AFS 98 mean grain size and its strongly monofractional composition (log W 37.0), i.e., 94% of the sand grains were distributed in only 2 fractions (Table 2). The 50% CB-145 hybrid blend adopted this more monofractional character to some extent and reflected the refracted shape characteristic of artificial sands, which are predominantly monofractional, in the resulting granularity curve (Figure 7a). Also, similar to 50% CB-95, the mixture showed lower 3-point bending strengths after curing compared to 50% CB-65, due to the larger surface area of fine grains that increase binder consumption. The resulting expansion of the mixture was the lowest (Figure 7b), and there would be a strong expectation that this mixture would also exhibit the highest resistance to fluidity. However, since monofractional sands are generally characterized by the greatest degree of linear dilation, and likewise perfectly round grains because of the greatest number of contact surfaces and thus the lowest possible porosity, this mixture experienced a sharp increase in volume during phase transformation, and the value of dilation thus obtained was almost unchanged by the mixture. Consequently, this was also reflected in the high occurrence of veinings and penetration.

The reason for the increased roughness of the cast surface of the 50% CB-65 and CB-95 hybrid mixes appears to be the almost perfect roundness of the CB sand grains used compared to silica sand, which, as a natural sand, is characterized by a more angular shape with rounded edges. The ideal round grains meet each other at the point of contact and therefore have minimal porosity after compaction. In contrast, angular grains contact each other in an area to produce higher porosity. In the case of hybrid mixtures, the round grain is unable to interact with the silica sand BG27 in a planar manner, so that for certain granulometry similar to silica sand, larger intergranular spaces are formed which, in the case of an untreated surface, allow the high convergence liquid metal (graphitising iron alloys) to follow the shape of these intergranular spaces, filling them and possibly penetrated the sand grains. This effect was observable in the 50% CB-65 and CB-95 samples, which showed, among other things, clogged sand grains and grain impressions, whereas this phenomenon did not occur in the 50% CB-145 sample (Figure 6). This is simply because the granulometry used and the grain fineness of the initial CB-145 helped to fill these inter-grain spaces in turn, but at the cost of a higher propensity to form veining.

The ideal condition, i.e., the complete prevention of the formation of veinings, and at the same time the maintenance of a surface with low roughness, was not achieved. The results indicate that hybrid mixtures with CB sand can be used to support the protective effect of the coating, when in the case of applying the coating on hybrid mixtures, only a minimal occurrence of veining and improved surface roughness can be expected. However, the results using a maximum of 50% CB were not sufficient for a complete replacement for the coating system.

In spite of initial expectations, the best results, i.e., limitation of veinings, were achieved with the 50% CB-65 hybrid mixture, mainly due to the granulometric composition suitably complementing the silica sand, which at the same time did not alter its polyfracturing or soften the grain size. However, it must be said that this result may not be generally applicable in all foundries, and simply adding CB-65 to any silica sand may not achieve such significant results. In addition to the granulometric composition, the shape of the grain, degree of compaction [13] and mineralogical purity are significant influences affecting the degree of dilation and the formation of veining. In particular, high SiO_2_ content in high-quality silica sands, together with different grain size, significantly increases dilatation, as demonstrated for example in this study [35]. It is then up to the individual to assess what grain size and quality of sand the given foundry uses, and whether it can accept the increased costs of a hybrid mixture. The application of the casting test and the use of a hybrid mixture adapted to specific conditions can nevertheless represent a suitable way to increase the surface quality of the castings.5. Conclusions

The aim of the experiment was to compare the quality of the cast surface of grey cast iron castings, which is influenced by the degree of thermal expansion of the sand core. Hybrid mixtures consisting of silica sand and Cerabeads artificial sand were created to reduce the linear dilation and thus reduce the risk of stress defects such as veining and surface roughness. By detailed analysis of the granulometric composition of the hybrid mixtures as well as the initial samples, determination of the dilatation behavior and 3-point bending strength of the fabricated cores, the causes of the formation of veining and increased roughness on the cast surface of the fabricated castings during the casting test were determined. It was possible to define these conclusions, which can further serve as a prediction of the potential risk of defects based on the chosen sand or its mixture:The tendency to dilatation and the formation of veining type defects is enhanced by the use of monofractional and round grained sandsThe propensity for veining increases the more the artificial sand mixed into the silica sand varies in terms of grain size and grading characteristics.A reduction in surface roughness can be achieved by using a mixture of sands where at least one sand contains very fine grains. On the other hand, there is an increased risk of lower core strength and, in the case of monofractional sand, also of greater dilatation and a higher risk of veining and penetration.A reduction of 52.9% in the occurrence of veining compared to the reference sample without coating (pure silica sand) and 42.5% compared to the reference sample with a coating and a positive effect on dilatation were achieved when using a 50% admixture of CB-65 artificial sand with granulometric properties closest to the initial silica sand.It was confirmed that a suitable composition of the hybrid mixture of sand can achieve a reduction in the occurrence of stress defects such as veining and penetration.

Using the same composition of sand mixture will not ensure defect reduction in general for all users. Due to the various influences, it is necessary to prepare the hybrid mixture completely individually.

## Figures and Tables

**Figure 1 materials-16-02004-f001:**
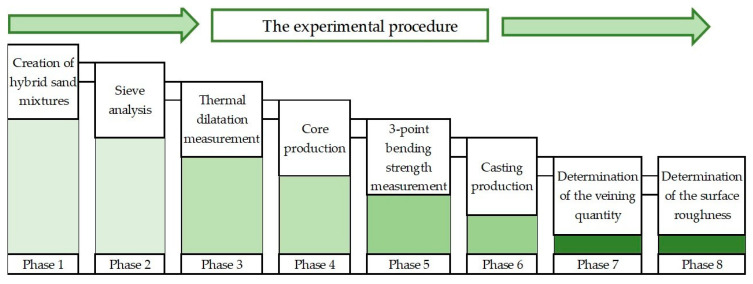
A schematic of measurement and sample production sequence.

**Figure 2 materials-16-02004-f002:**
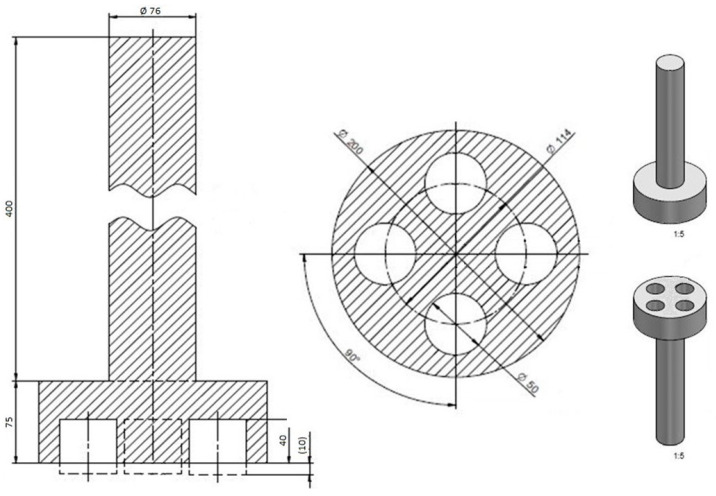
Schematic of the test casting model (own).

**Figure 3 materials-16-02004-f003:**
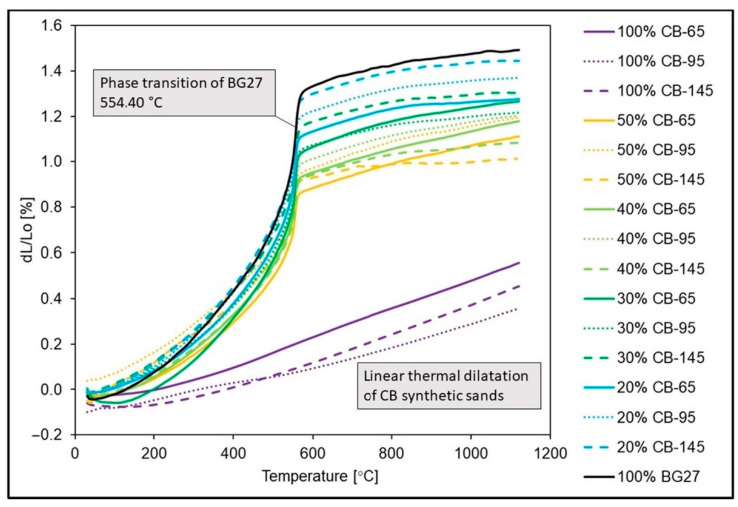
Dilatation curves for hybrid sands and reference samples of 100% pure sands.

**Figure 4 materials-16-02004-f004:**
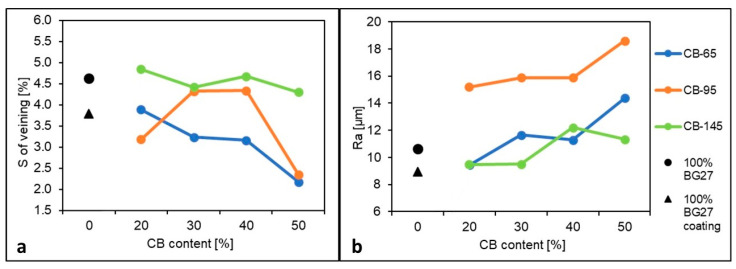
Casting test results: (**a**) resulting surface areas with veining and (**b**) roughness of the cast surface.

**Figure 5 materials-16-02004-f005:**
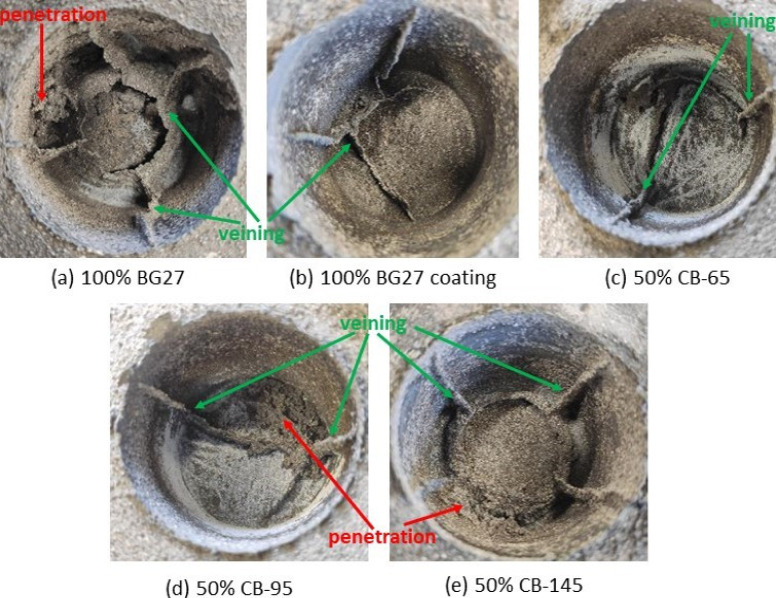
Appearance of surface defects such as veining and penetration on BG27 reference castings and castings with 50% CB addition.

**Figure 6 materials-16-02004-f006:**
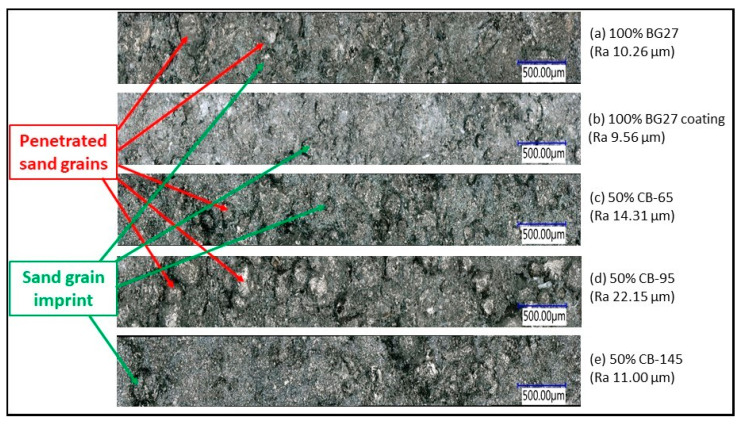
Cast surface roughness of concrete samples on BG27 reference castings and castings with 50% CB addition. The roughness Ra is given for the specific photograph of the specimen from which it was determined.

**Figure 7 materials-16-02004-f007:**
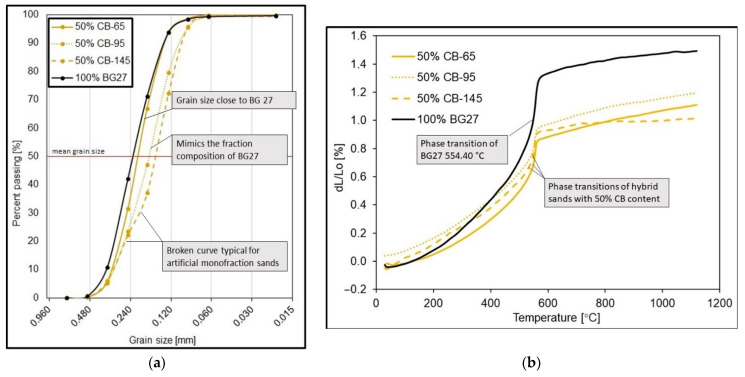
(**a**) Cumulative curves of granularity and (**b**) dilatation curves for 100% BG27 and hybrid sands with 50% CB content.

**Table 1 materials-16-02004-t001:** Marking of samples according to their composition.

Marking of A Sample	BG27	CB-65					CB-95					CB-145				
	100%	20%	30%	40%	50%	100%	20%	30%	40%	50%	100%	20%	30%	40%	50%	100%
CERABEADS (CB) [%]	0	20	30	40	50	100	20	30	40	50	100	20	30	40	50	100
Silica sand BG27 [%]	100	80	70	60	50	0	80	70	60	50	0	80	70	60	50	0

**Table 2 materials-16-02004-t002:** Parameters of tested hybrid sand mixtures and particle size distribution.

Sample/Content		Retained Part on Sieves [%]									Results	
		0.710	0.500	0.355	0.250	0.180	0.125	0.090	0.063	Pan	AFS [-]	log W [-]
BG27	100%	0.04	0.60	10.19	31.17	29.22	22.55	4.65	0.97	0.34	61	64.0
CB-65	20%	0.02	0.43	8.73	27.41	31.96	24.68	4.64	1.10	0.20	62	65.7
	30%	0.03	0.36	8.42	27.31	33.16	24.37	4.18	1.19	0.18	62	64.6
	40%	0.11	0.35	7.04	26.05	33.51	25.33	4.53	1.05	0.19	62	64.2
	50%	0.01	0.25	5.60	25.69	35.28	26.90	4.82	1.23	0.09	64	62.9
	100%	0.05	0.04	0.07	16.30	40.11	34.29	6.44	1.98	0.10	70	56.3
CB-95	20%	0.01	0.35	8.04	25.20	26.70	25.93	9.85	2.44	0.63	68	69.3
	30%	0.00	0.31	7.74	24.12	26.33	27.39	11.07	2.30	0.23	69	69.7
	40%	0.01	0.11	6.07	21.44	25.12	30.04	13.49	3.01	0.39	73	69.9
	50%	0.01	0.16	5.03	18.27	23.49	32.55	15.95	3.80	0.47	77	70.1
	100%	0.01	0.01	0.07	5.96	18.31	39.37	28.23	6.73	0.56	92	60.5
CB-145	20%	0.07	0.45	8.97	25.61	23.40	26.21	12.76	1.82	0.21	69	70.7
	30%	0.02	0.27	7.53	23.10	20.72	28.06	17.34	2.26	0.22	73	70.9
	40%	0.01	0.17	5.49	18.87	18.01	32.52	21.27	3.05	0.37	79	69.9
	50%	0.01	0.25	5.77	16.15	14.86	35.12	23.56	3.61	0.20	81	69.3
	100%	0.01	0.03	0.02	0.05	0.31	57.50	36.53	4.40	0.24	98	37.0

**Table 3 materials-16-02004-t003:** Thermal expansion results of hybrid mixtures and reference pure samples.

dL/Lo	BG27	CB-65					CB-95					CB-145				
	100%	20%	30%	40%	50%	100%	20%	30%	40%	50%	100%	20%	30%	40%	50%	100%
[%]	1.49	1.28	1.27	1.18	1.11	0.56	1.37	1.22	1.20	1.19	0.36	1.45	1.30	1.08	1.01	0.46
Sx [%]	0.182	0.076	0.102	0.033	0.077	0.031	0.038	0.137	0.068	0.117	0.023	0.016	0.136	0.086	0.303	0.173

**Table 4 materials-16-02004-t004:** Results of 3 point bending strength of hybrid sand compounds and the reference sample of pure silica sand.

σ	BG27	CB-65				CB-95				CB-145			
	100%	20%	30%	40%	50%	20%	30%	40%	50%	20%	30%	40%	50%
0 h [MPa]	4.00	3.38	3.74	3.69	3.25	2.84	3.07	2.86	3.02	2.84	2.74	2.84	3.05
24 h [MPa]	4.52	4.67	4.70	4.75	4.54	4.47	4.38	3.69	3.70	3.96	3.40	3.42	3.82
Increase [%]	13.0	38.2	25.7	28.7	39.7	57.4	42.7	29.0	22.5	39.4	24.1	20.4	25.4

**Table 5 materials-16-02004-t005:** Result of casting test—measured values of veining and roughness of the cast surface.

	BG27		CB-65				CB-95				CB-145			
	100%	100% Coating	20%	30%	40%	50%	20%	30%	40%	50%	20%	30%	40%	50%
S [mm^2^]	382.05	312.45	320.25	266.55	260.70	179.70	262.35	356.55	357.60	194.10	403.55	364.15	385.35	354.33
Sx [mm^2^]	38.10	131.57	4.25	7.45	5.30	16.70	25.25	35.35	38.5	2.30	3.95	36.15	10.15	20.48
S [%]	4.63	3.79	3.88	3.23	3.16	2.18	3.18	4.32	4.34	2.35	4.85	4.42	4.67	4.30
Sx [%]	0.46	1.60	0.05	0.09	0.06	0.20	0.31	0.43	0.47	0.03	0.00	0.44	0.12	0.25
Ra [μm]	10.65	8.93	9.45	11.65	11.30	14.37	15.19	15.87	15.89	18.61	9.46	9.50	12.20	11.34
Sx [μm]	1.66	1.67	3.27	1.75	4.14	1.58	2.52	3.61	3.04	1.73	0.74	3.33	3.50	0.73

## Data Availability

Not applicable.

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
