# Peer review of "Dilatation of New Progressive Hybrid Sand and Its Effect on Surface Structure, Roughness, and Veining Creation within Grey Cast Iron"

_materials, 2023, doi:10.3390/ma16052004_

Round 1
Reviewer 1 Report
Bašistová et al. have presented the manuscript titled: Dilatation of new progressive hybrid sand and its effect on surface structure, roughness and veining creation within grey cast iron. Overall presentation of the article is good. I have few suggestions for the authors about this article.
1. Abstract portion contains the long sentences of usually general discussion, this breaks the core concept what authors want to describe. I suggest the authors to rephrase the abstract with precise technical language to express their findings (result values).
2. Please change the caption of Table 1.
3. Method section line 121, “sands used are characterized by high chemical purity and similar bulk density…” what methods are used by the authors to evaluate the purity of the sand? And how they have measured the density? I suggest the authors to calculate the porosity of the samples as well.
4. I suggest the EDS measurement is important to confirm the purity or authors should go towards XRD at least.
5. As authors have added the roughness on the surface in the title, porosity and density plots are important.
6. Fatigue measurements like the exposure to water or open environment effect are important for this study.
7. In Figure 3, there is strong variation for the softening of sample CB-95, please provide the reason of such variation?
8. Mostly at 40%CB content there comes the change in trend of roughness and softening, is there some specific reason for that?
Author Response
Dear reviewer,
thank you for your comments and suggestions for improving our manuscript. Based on your recommendations, we've edited the manuscript as follows:
- Abstract portion contains the long sentences of usually general discussion, this breaks the core concept what authors want to describe. I suggest the authors to rephrase the abstract with precise technical language to express their findings (result values).
- Abstract has been modified according to suggestion.
- Please change the caption of Table 1.
- The caption of Table 1 has been changed.
- Method section line 121, “sands used are characterized by high chemical purity and similar bulk density…” what methods are used by the authors to evaluate the purity of the sand? And how they have measured the density? I suggest the authors to calculate the porosity of the samples as well.
- Thank you for your comment, the text has been edited. The chemical purity here referred only to silica sand and was given by the supplier. Information about supplier's declaration, as well as a study indicating the SiO2 content for Biala Góra silica sand, has been added to the text. In addition, general information including references regarding minimum SiO2 content for foundry silica sands has also been added to the text of chapter 2.1.
- Bulk density was performed during previous experiment. Information about resulted bulk density with refference was added to the text of chapter 2.1.
The methodology used: freely pouring the dried sample into a container of well-defined volume (graduated cylinder), where the sand sample was shaken to a constant height by vibration. The volume of the compacted sample was read using a ruler and the sample was weighed. The bulk density was then determined by the ratio of the sample weight to the actual sample volume in the container. At least 3 measurements were performed for each hybrid mixture.
- Porosity: very smooth core surfaces were achieved due to the fabrication of the cores using metal coreboxes. The surface defects (as deep penetration and veining) and increased roughness in the veining area investigated were related to the manifestation of the thermal dilation of the hybrid sand and the resulting braking stress, not to core porosity. For this reason, the calculation of core porosity was considered irrelevant to the experiment and was not included in the procedure.
- I suggest the EDS measurement is important to confirm the purity or authors should go towards XRD at least.
- It is true that the chemical purity of silica sand influences its thermal dilatation value, with high SiO2 content leading to increased dilatation. However, due to the mixing of silica sand with CERABEADS artificial sand with a different chemical composition (aluminosilicate) to reduce dilation and stress defects, the initial chemical purity of the silica sand was not so important for the experiment and we worked with the supplier's certificate.
Information about supplier's declaration, as well as a study indicating the SiO2 content for Biala Góra silica sand, has been added to the text of chapter 2.1.
- As authors have added the roughness on the surface in the title, porosity and density plots are important.
- Bulk density of hybrid mixtures was performed during previous experiment. Information about resulted bulk density with reference was added to the text of chapter 2.1.
- The surface roughness investigated was assessed in terms of stress defects caused by thermal expansion of the sand, where increased roughness can be the mildest manifestation and often accompanies veining. As increased surface roughness can also be a manifestation of overly coarse-grained sand, sparse (i.e. porous) cores, with liquid metal penetrating the intergranular spaces, an attempt was made to minimise this cause by producing cores by ramming into metal cores, whereby a highly dense and smooth surface layer is achieved. The use of a reference core coated with a protective coating was also adopted, which also serves as a protection against metal penetration into the sand and to increase the smoothness of the cast surface. For this reason, no assessment of the porosity of the core was undertaken.
The information has been added into the text of chapter 2.2.
- Fatigue measurements like the exposure to water or open environment effect are important for this study.
- Thank you for the suggestion. In the event that veining, penetrations or other stress defects appear, even in a small scale, on the available surfaces, these defects are repaired by extensive grinding. In that case, the surrounding surface layer and cast structure is removed. If these defects occur in inaccessible areas (internal surfaces of narrow cavities, e.g. valve casting), these defects cannot be effectively removed by grinding and the casting is discarded as a scrap piece. In both cases, knowledge and solutions to the fatigue behaviour of the cast surface in the area of these defects when exposed to water or environmental effects (e.g. corrosion rate) are then completely irrelevant. The object of the experiment was to avoid or reduce the occurrence of these defects in general. For this reason, this measurement was not included in the experiment.
The information has been added to the chapter 4.
- In Figure 3, there is strong variation for the softening of sample CB-95, please provide the reason of such variation?
- Thank you for your comment. After checking, an error was found in the rewriting of the numerical result. The increase in the occurrence of veining in the case of 40% CB-95 should not have been so high. Correction has been made in the text and graph.
Reasons for cracking of samples: in Figure 3 for CB-95 samples it is not softening of the samples, but cracking of the samples (cores) due to stress caused by the volumetric change of the cores during casting that allowed the metal to penetrate these cracks.
The addition of 20% CB-95 replicates the dilation pattern for a pure 100% BG27 sample, just with less dilation. With the gradual addition of CB-95, although the stresses decreased, the steepness of the volume change (steeper dilation curve with a steep increase from the beginning) increased during the β↔α SiO2 conversion around 573°C for the 30% and 40% CB-95 samples, as shown in the dilation graph. The 50% CB-95 sample exhibited the most gradual volume change, despite the fact that the resulting dilation value was almost identical to the 30% and 40% CB-95 samples. This is explained by the suppression of the typical manifestation of high SiO2 content under thermal loading, i.e., abrupt and sudden volume change, by the high degree of its replacement by sand with low dilatation. Here, although dilation was higher at low temperatures, it was gradual (more linear curve) and the volume change during phase transformation was not as high.
- Mostly at 40%CB content there comes the change in trend of roughness and softening, is there some specific reason for that?
- In the case of the addition of 40% CB, there was a shift of the temperature of the beginning of the phase transformation by several degrees higher (a little differently for each mixture) compared to the mixture of 20-30% CB, and the mixture retained a sharp volume change. In the case of 50% CB, this temperature was also shifted, but the volume change was slower and reached lower values at the phase transformation temperature. This is evident from the dilation graph.
We hope that you will find our edits to be a step towards improving the submitted paper.
Kind regards
Martina Bašistová

Reviewer 2 Report
Dear Author(s), the manuscript ‘Dilatation of new progressive hybrid sand and its effect on surface structure, roughness and veining creation within grey cast iron’, Manuscript ID: materials-2222792, have some crucial weakness that must be revised suitably.
Please find below some, of the most significant issues:
1. The sentence ‘The effect of the composition of a hybrid mixture of silica and artificial sand on reducing dilation and increasing the resistance to surface defects was demonstrated in an experiment. It was found that the use of 50% Cerabeads sand of the same granulometry as the initial silica sand is the key factor, which leads to a reduction in the incidence of veining by up to 52.9% without the use of a protective coating.’ Lines 16-20, should be reflected according to the results presented in the manuscript. In the current form, it looks like the proposal from the literature review than the newly applied.
2. I would suggest adding some additional keywords, like surface structure, and surface roughness.
3. The (main) aim of the work, mentioned comprehensively in lines 100-111, does not derive from the lack of the current state of knowledge. Straightly, there is a disproportion between the whole ‘Introduction’ section and the aim of the work.
4. The motivation of the work is not highlighted in the ‘Introduction’ section as well. All this makes this section, even suitably presented and well-written, weak and the improvement reasonably required.
5. The novelty should be received from the critical review of the literature, respectively, this second does not exist. From that matter, the identification of the novelty requirement is not obtained from the ‘Introduction’ section, unfortunately.
6. Section no.2, including three subsections, is presented comprehensively with the methods applied, nevertheless, I suggest adding some, even general, graphs with a presentation of the procedure including all of the experiments tested. Currently, the reader feels lost.
7. Referencing the sentence ‘The roughness of the cast surface was measured on the specimens thus prepared, the roughness values always being read from the centre of the surface.’, lines 218-220, some more words on the ‘centre of the surface’ should be provided. Respectively, why the edge-area were not studied? It must be justified.
8. According to the sentence ‘Thus, 4 values were obtained from each cavity along the core. In case a veining was present in the selected location for measurement, the roughness measurement was performed on the nearest surface next to the veining. Measurements were performed on a VHX-6000 digital optical microscope (Keyence) at 500x magnification.’ lines 220-224, including some response on the surface roughness measurement, the Author(s) must refer to both measurement uncertainty (repeatability) and measurement noise (errors), like:
(1) https://doi.org/10.1088/2051-672X/3/3/035004
(2) https://doi.org/10.3390/s22030791
(3) https://doi.org/10.1016/j.cirp.2014.03.086
9. Concerning section 3, for each of the subsections (3.1, 3.2 and 3.3) some, even 2-3 sentences, concluding the main (general) results, for each of the subsections, should be presented at the end of each of those sections parts. Currently, it is difficult to receive one, general result purpose from those subsections separately.
10. Some critical discussion in section 4 must be provided, respectively, the Author(s) should divide this section into two separate subsections to highlight both the advantages and disadvantages of the proposals. Currently, it looks like it is a perfect solution. Scientists should always provide some criticism of the work, especially presented by themselves.
11. Improve the ‘Conclusions’ section, the Author(s) should add some sentences for future prospects. In the current form, it looks like all of the results are completed. It, probably, is received by the lack of critical review in all of the sections (subsections), especially in the ‘Discussion’ one.
Moreover, two additional (editorial) modifications are required as well:
12. What is the e-mail address in line 7, respectively, ‘e-mail@e-mail.com’?
13. Sentences in lines 39, ‘Veining are most often found…’, and 41, ‘Veining are most commonly…’, should be re-written to avoid the repetition or, respectively, similarity.
14. The shortcut of the ‘Biała Góra’ (BG27) should be referenced, if exist previously.
15. The quality of Figures 2, 3 and 6 should be improved.
16. The ‘References’ should be provided with full DOI links, not only short forms, if exist.
From the above, the reviewed manuscript must be improved significantly before any further processing, if allowed by the Editor.
Author Response
Dear reviewer,
thank you for your comments and suggestions for improving our manuscript. Based on your recommendations, we've edited the manuscript as follows:
- The sentence ‘The effect of the composition of a hybrid mixture of silica and artificial sand on reducing dilation and increasing the resistance to surface defects was demonstrated in an experiment. It was found that the use of 50% Cerabeads sand of the same granulometry as the initial silica sand is the key factor, which leads to a reduction in the incidence of veining by up to 52.9% without the use of a protective coating.’ Lines 16-20, should be reflected according to the results presented in the manuscript. In the current form, it looks like the proposal from the literature review than the newly applied.
- - Abstract has been modified according to suggestion.
- I would suggest adding some additional keywords, like surface structure, and surface roughness.
- Another keywords have been added as suggested
- The (main) aim of the work, mentioned comprehensively in lines 100-111, does not derive from the lack of the current state of knowledge. Straightly, there is a disproportion between the whole ‘Introduction’ section and the aim of the work.
- The aim of the work has been rewritten.
- The motivation of the work is not highlighted in the ‘Introduction’ section as well. All this makes this section, even suitably presented and well-written, weak and the improvement reasonably required.
- The motivation of the work has been rewritten together with previous suggestion.
- The novelty should be received from the critical review of the literature, respectively, this second does not exist. From that matter, the identification of the novelty requirement is not obtained from the ‘Introduction’ section, unfortunately.
- The aim, motivation and novelty of the work has been rewritten as suggested.
- Section no.2, including three subsections, is presented comprehensively with the methods applied, nevertheless, I suggest adding some, even general, graphs with a presentation of the procedure including all of the experiments tested. Currently, the reader feels lost.
- Schema of sequence of individual measurements has been added into chapter 2 as Figure 1.
- Referencing the sentence ‘The roughness of the cast surface was measured on the specimens thus prepared, the roughness values always being read from the centre of the surface.’, lines 218-220, some more words on the ‘centre of the surface’ should be provided. Respectively, why the edge-area were not studied? It must be justified.
- The roughness was measured if possible in the areas near defects and in the center of the surface. The center of the casting was chosen with regard to the possible presence of impurities in the upper and lower parts of the casting and also because of the possible increased occurrence of baked-on sand near the detected defects.
The information has been added into the text – chapter 2.4.
- According to the sentence ‘Thus, 4 values were obtained from each cavity along the core. In case a veining was present in the selected location for measurement, the roughness measurement was performed on the nearest surface next to the veining. Measurements were performed on a VHX-6000 digital optical microscope (Keyence) at 500x magnification.’ lines 220-224, including some response on the surface roughness measurement, the Author(s) must refer to both measurement uncertainty (repeatability) and measurement noise (errors), like:
(1) https://doi.org/10.1088/2051-672X/3/3/035004
(2) https://doi.org/10.3390/s22030791
(3) https://doi.org/10.1016/j.cirp.2014.03.086
- On each surface were performed 5 separate measurements. In total 20 separate measurements were done for each cavity (4 surfaces x 5 measurements. The results were then averaged. In case of error measurement, the comparative measurement was done. Information has been added into the text of chapter 2.4.
- Concerning section 3, for each of the subsections (3.1, 3.2 and 3.3) some, even 2-3 sentences, concluding the main (general) results, for each of the subsections, should be presented at the end of each of those sections parts. Currently, it is difficult to receive one, general result purpose from those subsections separately.
- Summaries of individual chapters have been added to the text as suggested.
- Some critical discussion in section 4 must be provided, respectively, the Author(s) should divide this section into two separate subsections to highlight both the advantages and disadvantages of the proposals. Currently, it looks like it is a perfect solution. Scientists should always provide some criticism of the work, especially presented by themselves.
- Discussion has been improved with references as suggested.
- Improve the ‘Conclusions’ section, the Author(s) should add some sentences for future prospects. In the current form, it looks like all of the results are completed. It, probably, is received by the lack of critical review in all of the sections (subsections), especially in the ‘Discussion’ one.
Moreover, two additional (editorial) modifications are required as well:
- Conclusion has been improved with references as suggested.
- What is the e-mail address in line 7, respectively, ‘e-mail@e-mail.com’?
- Correct e-mails have been added.
- Sentences in lines 39, ‘Veining are most often found…’, and 41, ‘Veining are most commonly…’, should be re-written to avoid the repetition or, respectively, similarity.
- The text has been corrected according to suggestion.
- The shortcut of the ‘Biała Góra’ (BG27) should be referenced, if exist previously.
- The shortcut of the „Biała Góra“ the BG27 is the designation given by the supplier for the Czech market (SAND TEAM). A reference to an earlier study where this shortcut was used (in accordance to supplier´s approval) has been added to the text.
- The quality of Figures 2, 3 and 6 should be improved.
- The Figures 2, 3 and 6 have been improved.
- The ‘References’ should be provided with full DOI links, not only short forms, if exist.
- DOI´s have been added where possible.
We hope that you will find our edits to be a step towards improving the submitted paper.
Kind regards
Martina Bašistová

Reviewer 3 Report
The authors in the work assessed the dilatation of new progressive hybrid sand and its effect on surface structure, roughness and veining creation within grey cast iron. The aim of the experiment was to compare the quality of the cast surface of grey cast iron castings, which is influenced by the degree of thermal expansion of the sand core. Hybrid mixtures consisting of silica sand and Cerabeads artificial sand were created to reduce the linear dilation and thus reduce the risk of stress defects such as veining and surface roughness. The authors have made an analysis of the granulometric composition of the hybrid mixtures as well as the initial samples, determination of the dilatation behavior and point bending strength of the fabricated cores, the causes of the formation of veining and increased roughness on the cast surface of the fabricated castings during the casting test were determined.
The article is very interesting, it raises a number of problems resulting from the casting production processes. As a reviewer of this work, however, I believe that the reviewed work requires many corrections, which will undoubtedly improve its quality. There is chaos in the work, the discussion is not a discussion but an analysis of the results (no comparisons to other works). There are a lot of errors:
1. Line 135 - title in English;
2. Lines 226-228 - this is probably a mistake;
3. Line 281- table 3. What is Sx?
Author Response
Dear reviewer,
thank you for your comments and suggestions for improving our manuscript. Based on your recommendations, we've edited the manuscript as follows:
Discussion has been improved with references as suggested.
- Line 135 - title in English;
- The title has been corrected
- Lines 226-228 - this is probably a mistake;
- Thank you for mentioning. It was a mistake. The text has been removed.
- Line 281- table 3. What is Sx?
- Sx is standard deviation. The standard deviations were accidently deleted during formatting of the text without noticing. The missing values were added into the Table 3. We very regret this mistake.
We hope that you will find our edits to be a step towards improving the submitted paper.
Kind regards
Martina Bašistová

Round 2
Reviewer 1 Report
Authors have revised the manuscript very well according to my suggestions. I suggest this manuscript can be published now in the journal.
Thank you
Reviewer 2 Report
Dear Authors, the manuscript titled 'Dilatation of new progressive hybrid sand and its effect on surface structure, roughness and veining creation within grey cast iron', Manuscript ID: materials-222279', was improved suitably, all of the raised issues were responded suitably so, respectively, in my opinion, the revised version of paper can be considered for publication in the current form in the Materials journal.
Reviewer 3 Report
The authors of the article corrected errors and responded to the reviewer's comments. In the article presented for review, the authors described their research very meticulously and with great care. All of the results contained in it were presented in a clear and legible way for the recipient. The conclusions and the study summary are consistent and well-formulated. Summing up, the reviewed work presents a high substantive and experimental value.